# Landscapes of the Yazd-Ardakan Plain (Iran) and the Assessment of Geotourism—Contribution to the Promotion and Practice of Geotourism and Ecotourism

Iulian Dincă [1,*], Seyede Razieh Keshavarz [2] and Seyed Ali Almodaresi [3]

1 Department of Geography, Tourism and Territorial Planning, University of Oradea, 410087 Oradea, Romania
2 Geography Department, Islamic Azad University, Yazd Branch, Yazd 89168 71967, Iran; keshavarz@iauyazd.ac.ir
3 GIS & RS Department, Islamic Azad University, Yazd Branch, Yazd 89168 71967, Iran; almodaresi@iauyazd.ac.ir
* Correspondence: idinca@uoradea.ro or iulian_dinca@yahoo.co.uk

**Abstract:** The attractions and capabilities of geomorphosites are among the unique assets of each country, and their identification, classification, and planning have great importance for the development of tourism. The purpose of this research was to identify, quantitatively analyse, and classify landforms treated as geosites and landscapes of Yazd Province, using the Pralong method. The present study is a descriptive analytical research based on library studies and field surveys. After determining the study area using satellite imagery, six geomorphosites were selected. Using the Pralong method, six geomorphosites were evaluated: Chak Chak, Mountain Eagle, Siahkooh, Shirkooh, Qanat, and Barfkhane Tezerjan. In this method, using the extraction of collected data from the questionnaires completed by 41 geotourism experts in the first stage, the four grades that were evaluated in terms of potential capability of geomorphosites include the appearance aesthetic, scientific, historical-cultural, and socio-economic variables, and in the next step, two variables of productivity value and quality of productivity were evaluated. The results of evaluations showed that the geomorphosites Chak Chak and Barfkhane Tezerjan had the highest score (0.62) and are the best geomorphosites for converting to geotourism and ecotourism applications, and they have high potential for attracting tourists. The second place was dedicated to the Shirkooh geomorphosite, which is considerable in terms of the average tourism grade. The best average productivity value belonged to the Chak Chak geomorphosite with 0.52, and the last one was Mountain Eagle with 0.32. All six attractions in this study had a medium grade in the qualitative scale index, meaning that they have good potential in geotouristic and ecotouristic points of view and could be improved by regional planning. Therefore, the results of this study can be used by local managers and planners to develop and promote geotourism and ecotourism.

**Keywords:** geosite; geomorphosite; Pralong method; geological and geographical diversity; identity characters

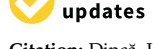



## 1. Introduction

Iran has a rich culture and civilization as well as a spectacular natural environment. Its natural and cultural diversity has promoted it to one of the leading countries in world tourism, and its archaeological, cultural, and natural attractions provide an excellent foundation for the development of geotourism and ecotourism. While Iran has a diverse range of geological phenomena, geotourism is emerging and forming [1].

The development of geomorphotourism is one of the ideas behind using landscape touring (or more frequently, landscape tours) [2], starting from the principle that the Earth's diversity at the ecological level is expressed also by the existence of geosites with historical-cultural, scientific, economic-social, visual, aesthetic [3], and even geo-aesthetic values [4].

From here to imaginary tourism at the expense of landscapes [5], it is a simple decision and cultural sense.

Landscape touring is highly regarded, particularly in industrialised countries. For ecotourism, in addition to the components belonging to natural landscapes, the components belonging to subnatural and humanised landscapes also count.

Geotourism was first defined from an academic perspective in 1995 by Hose, quoted by [6]. Generally, geotourism refers to a collection of activities, infrastructure, and services aimed at enhancing the value of geological heritage through tourism [6]. In fact, geotourism is one of the fields that addresses tourism-related studies that focus on the investigation of geosites dominated by geological and geomorphological perspectives. The term geomorphosite was first proposed in 1993 by Panizza, quoted by [7]. Various terms were used to describe the concept of geomorphological heritage, including geomorphological assets, geomorphological goods, geomorphological sites, geomorphological geotopes, and geomorphological sites of interest [8]. Geotourism may be further described as having a number of basic characteristics. There are five key principles that are fundamental to geotourism: It (a) is geologically based (that is, based on the earth's geoheritage), (b) is sustainable (i.e., economically viable, community enhancing, and it fosters geoconservation), (c) is educative (achieved through geo-interpretation), (d) is locally beneficial, and (e) generates tourist satisfaction [7].

Modern research [9,10] provided one of the most common but explicit definitions of geotourism. They pointed out that geotourism is a type of tourism in a natural area that focuses specifically on geology and landscape [10] or was reinterpreted as an approach to studying landscape–tourism interactions [11]. It promotes tourism to sites and the protection of geographical diversity and the understanding of earth sciences through appreciation and learning. This is achieved through independent visits to geological features, the use of geographical routes and views, guided tours, geographical activities, and the support of geosite visitor centres [12], among many others, such as signed geotrails, interpretative panels, geoguides, and so on.

Geomorphosites/geosites as landscapes [13] are considered natural goods not only due to their intrinsic values [14] (scientific, aesthetic) but also due to their external values (ecological, historical, cultural, economic). Thus, they are the primary drivers of geotourism development [8]. Ecotourism is a form of nature-based tourism that contributes to social and environmental wellbeing. It is also referred to as green tourism due to its environmentally friendly and educational nature [15]. When developed sustainably, ecotourism has the potential to mitigate negative impacts of tourism, enhance cultural and environmental integrity, improve resource management, and generate revenue [16]. There are views that discuss ecotourism in terms of sustainability that are based on ecological, economic, and socio-cultural pillars [17]; on the natural and social environment [18]; on conservation perspectives and marketing strategies [19]; on plans, including sustainable agriculture, micro-industry, and other activities [20], cited by [18]. However, the representation of ecotourism can go beyond the delicate conceptual 'lock-in' established by the 'form of nature-based tourism'. This is because ecotourism attracts tourists who appreciate not only the natural environment but also the social and cultural environment. In inhabited rural areas, a number of ecotourists interested in discovering nature seek to explore in detail including the anthropo-cultural attractions of the places. Where, at a certain distance from these rural areas [21–24], there is a natural and attractive heritage, ecotourists have the opportunity to expand their knowledge of such attractions by exploring this heritage on the basis of valuable or at least interesting geological, petrographic, and geomorphological features. In other words, these visitors find that they meet the conditions for typical geotourism activities. In this way, the two types of tourism, ecotourism and geotourism, find common elements and similarities in terms of concept and practice, falling under the forms of rural tourism/agrotourism, cultural-historical tourism, and the tourism of visit-discovery.

In this way of expressing the desire of ecotourists to know, explore, and discover, ecotourism has the potential to be a significant means of poverty alleviation in economically disadvantaged areas endowed with significant natural resources [25]. Thus, ecotourism is a much more complex form of tourism than is being made out.

The two forms of tourism, ecotourism and geotourism, are considered forms of experiential tourism [26], and they sometimes also raise questions such as about the differences between them in terms of their elements of attraction. Ecotourists, in addition to ecological activities [27] related to natural attractions, add and focus seriously on the cultural side of the wild side, but also on the constructed, humanised, and culturalised sides of the countryside outside or close to purely geotouristic attractions (hence the assimilation of ecotourism with rural tourism or agrotourism). It also sets out the elements of closeness between geotourism and ecotourism, both of which are designated by the need to explore natural, pristine places without human intervention that provide lasting experiences [28]. In other words, geotourism is 'twinned', as it is complementary to ecotourism. Likewise, geotourism can also be said to be a separate type of ecotourism, with some voices differentiating geotourism from ecotourism in that geotourism focuses on the working landscape of the region [29].

The general working hypothesis is the interdisciplinary treatment of natural and anthropogenic resources in the Yazd-Ardakan Plain of Iran, understanding by resources the geomorphosites/geosites and landscapes of the investigated unit. The operational working hypothesis is to search for the conditional and determining relationship between the couple of geological substratum diversity–landforms–landscapes and the creation of the appropriate framework for the manifestation and promotion of local geotourism and ecotourism. Working variables include geomorphosite/geosite assessment and thematic landscape analysis.

Through interdisciplinarity [30], the meaning pursued in this study was to arrive at as fine an understanding as possible, as close as possible to the unsophisticated judgement of tourists (be they geotourists or ecotourists) about the landscapes they visit and with which they immediately socialize and encounter.

## 2. Research Background

The central pillar of the study is the interest in the landscape heritage of the study area. These landscapes are organised into components that are common to both geotourism (particularly the geological and geomorphological environment, i.e., geolandscapes [31]) and ecotourism (adding abiotic, biotic, and cultural components) (e.g., ecotourism landscapes—[32]).

Geotourism and ecotourism are relatively new concepts in tourism studies, but they have grown in popularity and prominence over the last few decades [12,33]. Accordingly, much research has been conducted in this regard. Of particular interest in this study is the pursuit of the benefits of transdisciplinarity through which local or indigenous ecotourism is conducted [16], the importance of ecotourism in the sustainable development of thematic products, and activities belonging to geoparks [34]. In other studies, geotourism is based on geology and landscapes [10,35], including features of natural, subnatural, and humanised landscapes [36,37]. Landscapes themselves become territorial resources for tourism [38,39], promoting the image of the countryside [40,41] and conservation [42–45], geotourism and recreation activities, and preservation [31,46,47]. Geotourism and ecotourism merge thematically, are close and complementary, and are types of tourism that highlight and orient tourists towards non-consumptive activities of biotic and abiotic wilderness [5,48,49] that should not be degraded by visiting and exploring. This form of tourism, ecotourism, is often associated with geotourism [10,17,18,50–53]. Here are two formulations in which reference was made to the equivalence, even belonging, of geotourism to ecotourism: "Geotourism is ecotourism with an added geological theme." [51] (p. 1); "Geotourism has great potential as a new niche ecotourism product, ... " [51] (p. 1).

All these parts of nature in which geotourism and ecotourism are or can be practiced are unmistakable landscapes (desert landscapes) [54] that are the object of tourist interest,

including photography sessions, with openings for cultural-aesthetic acquisitions and advanced qualitative experiences.

Research in the Iranian geographical area has produced notable results related to geotourism, including ecotourism, which is similar and complementary to geotourism. The intention of the research was that, due to the high geodiversity, biodiversity, and numerous historical and cultural attractions, as many geoparks as possible should be proposed and accepted [52,55,56].

Related to the arid landscapes of the Yazd-Ardakan Plain are recommended studies of geotourism attractions [57] and management solutions to control aridization [58]. The results showed that very beautiful and unique desert attractions, saline lands, Zoroastrian temples located on the slopes of high mountains, aqueducts, and other landscapes provide a suitable ground for attracting scientific tourists. Rezaei [59] conducted a study in Yazd city, examining residents' perceptions of the tourist impact.

Farsani et al. [60] investigated tourists' satisfaction and motivation with Isfahan mining geotours. The results indicated that tourists were interested in discovering new destinations, as well as staying in geo-accommodations. They also indicated that visiting underground and surface mining operations, as well as participating in geo-sports, were among their geotour priorities.

Other researchers [61,62] conducted studies on the role of environmental education in geotourism destinations and the significance of urban geomorphological heritage for urban geotourism development. The first study indicated that the lowest score was related to environmental knowledge and facilities. As a result, it was necessary to improve the education of tourists with limited environmental knowledge in order to increase their environmental awareness. The results of the second study indicated that, of the 32 geomorphosites inventoried in the karstic, fluvial, tectonic, anthropogenic, and specific geomorphosite categories, the Falak-ol-Aflak Castle Hill received the highest scores in all three scientific, educational, and geotourism criteria.

Additionally, Ranjbaran et al. [63] examined the geotourism attraction of Hormuz Island in their research. The study concentrated on fieldwork, which included data collection and photography, as well as a review of previously published articles and books. Due to geotourism's primary attractions, such as rocky beaches, sea caves, vibrant salt domes, and coral reefs, it was demonstrated that Hormuz Island can be proposed and exploited as a geopark [64]. The Pralong method [65] is used in a number of interesting studies. The Pralong method, although not applied today by all researchers assessing the tourism potential of geosites, is still of wide interest. This is explained by the fact that the method uses the six indicators, the experience and results not only of the method's proponent, but also the experience and results of other studies published before Pralong. In addition, the benchmark indicators taken into consideration in this study fit very well with our intentions of multi- and interdisciplinary analysis and thematics. In other words, the first four indicators gather the convergent interest for the component part and the socio-economic-cultural relevance of geosites, and the last two indicators outline the openness for the valorisation of the same geosites. Perhaps only the economic indicator carries some critical discussion, in the sense that subjectivity factors into what is meant by the economic value of geosites. It may be irrational rapid economic exploitation that carries risks for the conservation of geosites, but we can also think of long-term economic exploitation, where financial benefits come in tandem with care for the existence and condition of the same geosites.

Amiri et al. [66] used the Pralong method to examine the Haraz Watershed's landform potential for educational purposes as well as tourist attraction. Similarly, Baboli Mokher and Ramesht [67] adopted the model to assess the geotouristic potential of the Tashan region of Behbahan city, in a quest to achieve sustainable development. Due to its ancient monuments and unique historical location, the historical area of Tashan (Kalgahzar) had the highest potential for developing tourism and attracting tourists in the region, earning a score of 0.62 for tourism and 0.46 for average value of productivity. Applying the same

method, Artugyan [68] appraised geomorphosites in karst terrains in Romanian's Banat Mountains, focusing on springs, caves, straits, and plateaus. The results demonstrated that it is critical to properly exploit these geomorphosites in the Anina karst area in order to protect the karst landscape.

Wondirad et al. [69] examined the stakeholder collaboration as a major factor for ecotourism development in developing countries. The findings of the study indicated that ecotourism stakeholders have ineffective interactions and collaborations. As a result, in under-resourced and remote destinations, the failure to empower and engage communities undermines ecotourism and jeopardizes the ecosystems' and communities' long-term survival.

Ching et al. [70] demonstrated that the opportunities and strengths associated with sustainable ecotourism development in Malaysia's Cameron Highlands outweigh the threats and weaknesses. They also established that mountainous areas in the region have a high capacity to become attractive ecotourism destinations.

Numerous research studies in the literature put forward ideas that highlight more or less similar visions. Our comprehensive vision of both forms of tourism, geotourism and ecotourism, is not necessarily more permissive, but it confirms, beyond the respect for nature, the condition of geotourists and ecotourists as the most profound connoisseurs of nature; however, it may appear, without necessarily looking for exceptional attractions, able and with maximum openness for the most active and energetic tourist services, as they are the closest by knowledge and education to the mysteries of each thematic area [71].

## 3. Materials and Methods

In this study, three-stage methods were applied. The research is captured schematically in Figure 1. The first stage of the methods applied included fieldwork, the completion of 41 questionnaires by Iranian experts with different specialisations (geography, geology, remote sensing, tourism, and urban planning), and their processing, as well as the application of the Pralong method. The Pralong method allowed the assessment of the geosites' potential for geotourism in the investigated area. In the third stage, these results on geosites were cross-disciplinarily corroborated on the basis of landscape studies, as geosites are intrinsically linked to landscape units. This last level of investigation is also linked to the identification of the practical and promotional aspects of geotourism and ecotourism, with these two forms of tourism exploiting the potential offered by landscape inventory.

The Pralong method was applied for the quantitative and qualitative analysis of geosites. With the Pralong method, the tourism value of every site is determined by the average of the four indicators of apparent beauty and scientific, historical-cultural, and socioeconomic aspects, which are scored from five different levels. In this method, the value of the current productivity of the sites was evaluated. In other words, productivity and product quality were used to assess the productivity value of geomorphosites to identify the potential and actual capabilities of the sites. The investigation of geosites and the broad scientific interest in them coagulate as a result of features of detail that are not only geological and geographical but are also hydrological, climatic, biological, and anthropogenic, all subordinate to the idea of time and the effects of time on geosites [14,72–75]. Time must be understood from a double perspective. The first refers to domain-specific quantitative measurements of the evolutionary dynamics and configuration of geosites (geology, biogeography, hydrology, climatology, habitat, population, and village culture). The other concerns the effects of the evolutionary dynamics and configuration of geosites on the aesthetics of the whole and of detail, which is visually received and emotionally consumed by visitors to the whole, understood as geosite. Geosites inevitably receive the effects of some of the most disruptive natural or social phenomena on the equilibrium state of those geosites. In this case, the most appropriate geoconservation management strategies need to be applied.

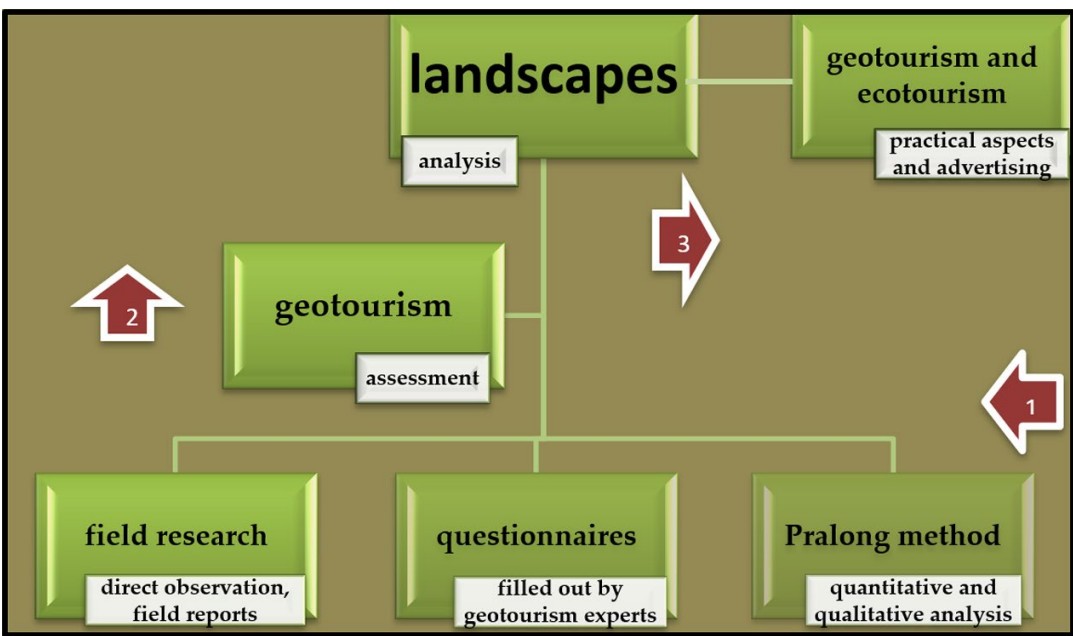

**Figure 1.** Research methodology scheme.

The value of geosites will be even better understood by passing through the filter of landscape heritage analysis, using the service of the visual design elements method, even if it must be agreed that there is subjectivity on this analytical route [76–79]. The methods proposed by Jakel and Bell used in this study incorporate a certain degree of subjectivism, but one that is unanimously accepted in landscape science. Although these methods emphasize the design of form and function of landscapes that relate to geosites, they are the ones that leave behind the sometimes exaggerated academism of classic non-landscape methods and shift the analytical meaning to the narrative of the elementary and practical understanding of landscapes by visitors and tourists.

In landscape science, the geosite is only the physical, material, and heritage part of a geographical space, and its landscapes are those that enhance the complexity and attractiveness of the geosite through its features. Finally, the value of the geosites and landscapes results in a potential tourist attraction that recommends and certifies the entire region for tourism. These values are based on the relevance of the natural, subnatural, and anthropogenic components of the area. They are reflected in the identity of the structure, organisation, and functioning of the landscapes that are the subject of geotourism and ecotourism attractions.

From our point of view, by bringing the theme of landscape to the forefront and linking it to geo- and ecotourism, we believe that the most appropriate formula for geo-diversity and ecodiversity knowledge in one place or another, for supporting a healthy civic education aimed at the sustainable protection of pure nature or humanised nature, is reached. The sense sought was to reach a fine-grained understanding as close as possible to the unsophisticated judgement of tourists (be they geotourists or ecotourists) about the landscapes they visit and with which they immediately socialize and confront. Tourists need to come to a simple understanding of landscapes, without the pretensions of scientists. In this way, we can be sure that the same landscapes will be responsible for a high level of awareness for tourists in the form of emotional satisfaction and cultural fulfilment, which geotourism and ecotourism devote to them.

### 3.1. Area of Study

Yazd-Ardakan Plain is located in the central part of Iran's Central Plateau (Figure 2), between latitude 29°52′ to 33°27′ north and 52°55′ to 56°37′ east longitude. The altitudes

of the area vary from about 666 m above the Azad Sea level (Rigzerin Desert near Aqda) to 4075 m (Shirkoh Peak) [58].

Yazd, which lies roughly in the centre of the studied unit, is one of the most important tourist cities in Iran, one of the oldest cities in the world, dating back to 3000 BC, and also the oldest city of clay in the Islamic world. Due to the increase in foreign tourists visiting this city in recent years, significant changes have occurred in the city's historical district [59]. Shirkoh Mountain is considered one of the scattered mountains of Central Iran, and it rises similarly to a high wall in the south and southwest parts of Yazd-Ardakan Province with a northwest–southeast trend. This mountain ranges from the east to Bohruk Mountain, Ibrahim Abad plain, and Mehriz city; from the south to Tang Chenar village, Degh Ernan, Nir, and Turan Pasht; from the west to the Godar Rigyuk region, the Ali Abad Damak district, and the cities of Islamia and Taft; and from the north it is limited by a short distance to the city of Yazd.

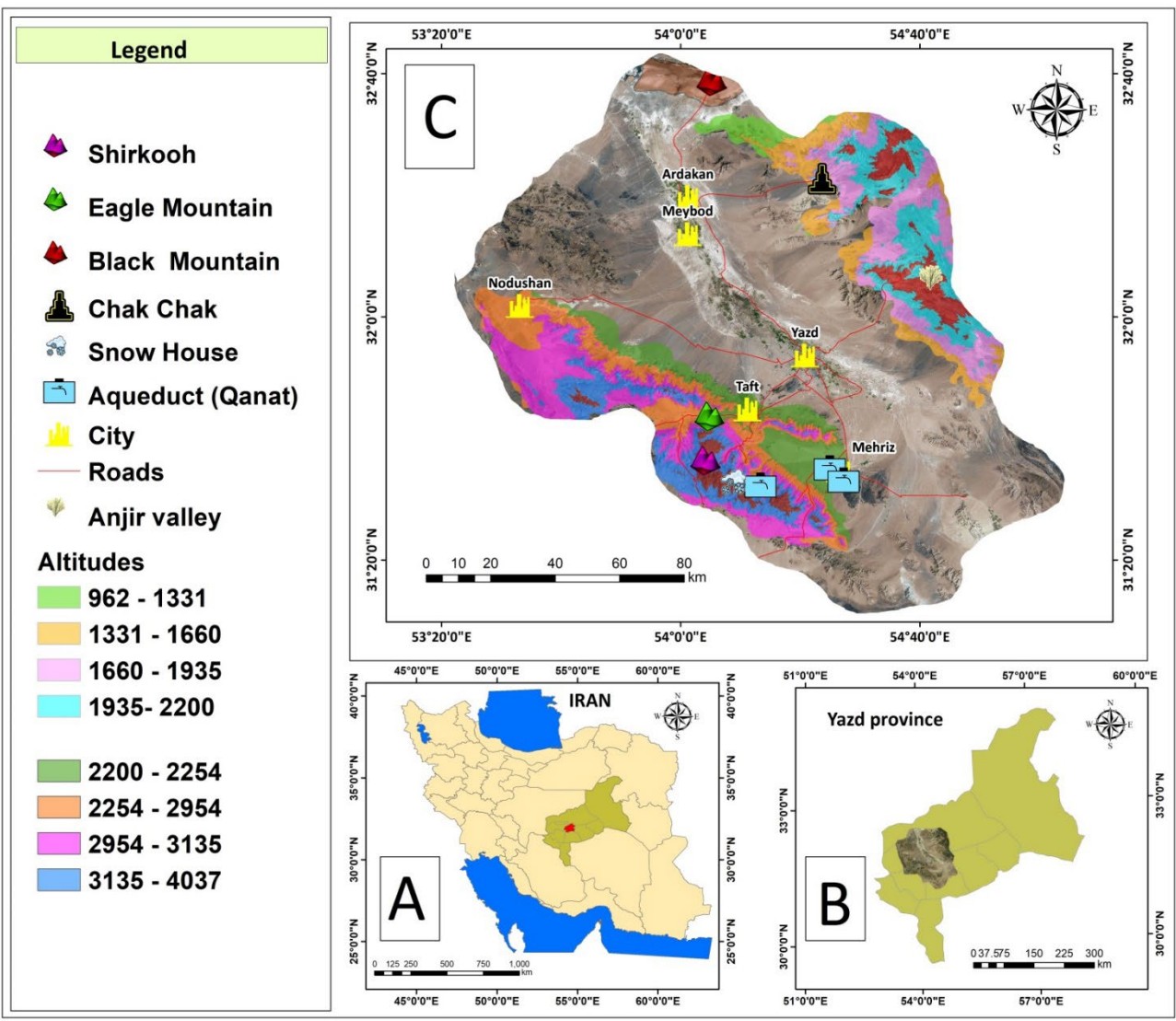

**Figure 2.** The geographical position of geotourism and ecotourism attractions in the Yazd–Ardakan Plain: location of Yazd Province in Iran (**A**,**B**); satellite image with the spatial distribution of geotourism and ecotourism attractions (**C**).

### 3.2. Geosites, Geomorphosites, Geotourism, and Ecotourism of Yazd-Ardakan Plain

Yazd Province's climatic conditions have left the majority of areas desolate and barren. The current ambience in this province is rooted in the ancient geological history of Iran

and the world. Only a few geotourism and ecotourism attractions, such as deserts, salt marshes (salt domes), sand dunes, aqueducts, glaciers, springs, karst caves, and clots, can be found in close proximity to one another in other parts of the world. With such a wealth of geotourism and ecotourism resources, this province can claim a distinct position among ecotourism and geotourism destinations, and the development of ecotourism will result in the socioeconomic progress of Yazd Province [57]. From Shirkooh Peak to Siahkooh Playa, we can see how all of the natural phenomena and landscapes, such as Tezerjan's mountain glacier, glacial moraine, alluvial plains, sand dunes, and playa, have been clustered together [57]. The most interesting landscapes relate in particular to the geosites marginally arranged on the more solid, mountainous structures located on the south-western and north-eastern sides of the Yazd-Ardakan Plain. The variety of geomorphological phenomena and other natural landscapes that are concentrated within short distances, as well as the presence of many very interesting and beautiful perspectives, attract geo- and ecotourists (Figure 3).

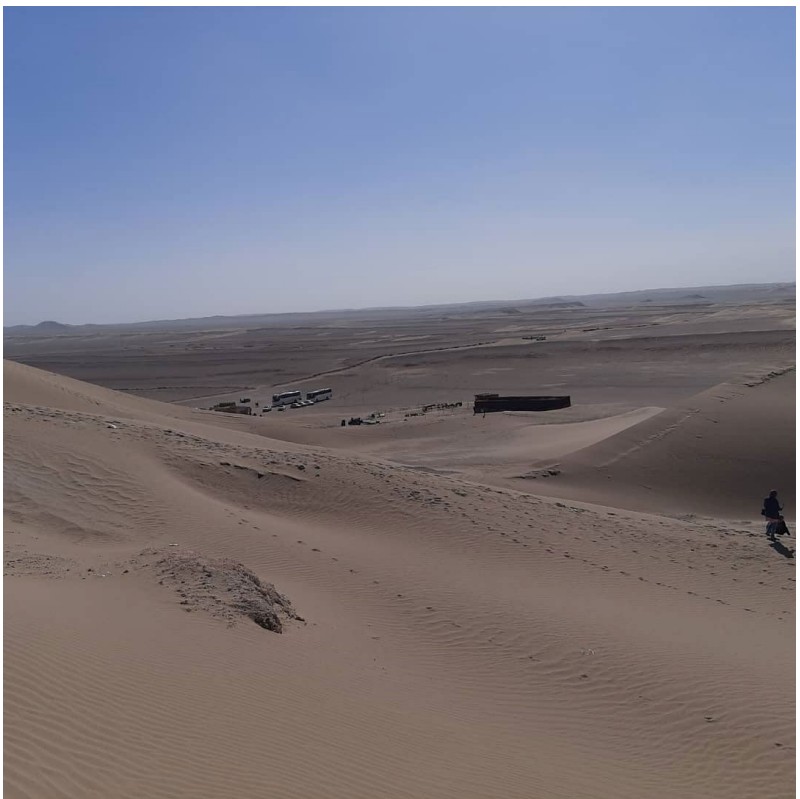

**Figure 3.** An example of a sightseeing, discovery, and socializing tourist activity with the impressive landscapes of large sandy accumulations and eco-camp facilities, part of the Black Mountain geosite (Photo source: J. Gorji®).

This is also the reason why most of the geosites that were evaluated in this study by the Pralong method belong to mountain landscapes. The central axis of the studied unit (Figure 2), designated by a broad, low, arid unit with soft alluvial sedimentary deposits, is less attractive from the point of view of landscapes, but it engages geo- and ecotourists in the casual activities typical of humanised landscapes (Figure 3). The Chak Chak shrine has resulted in an increase in religious tourism, which is a significant source of revenue and a significant driver of Ardekan and Yazd's economic development.

The general geological and geographical characteristics of all geomorphosites studied can be viewed and are analysed below (Figure 4).

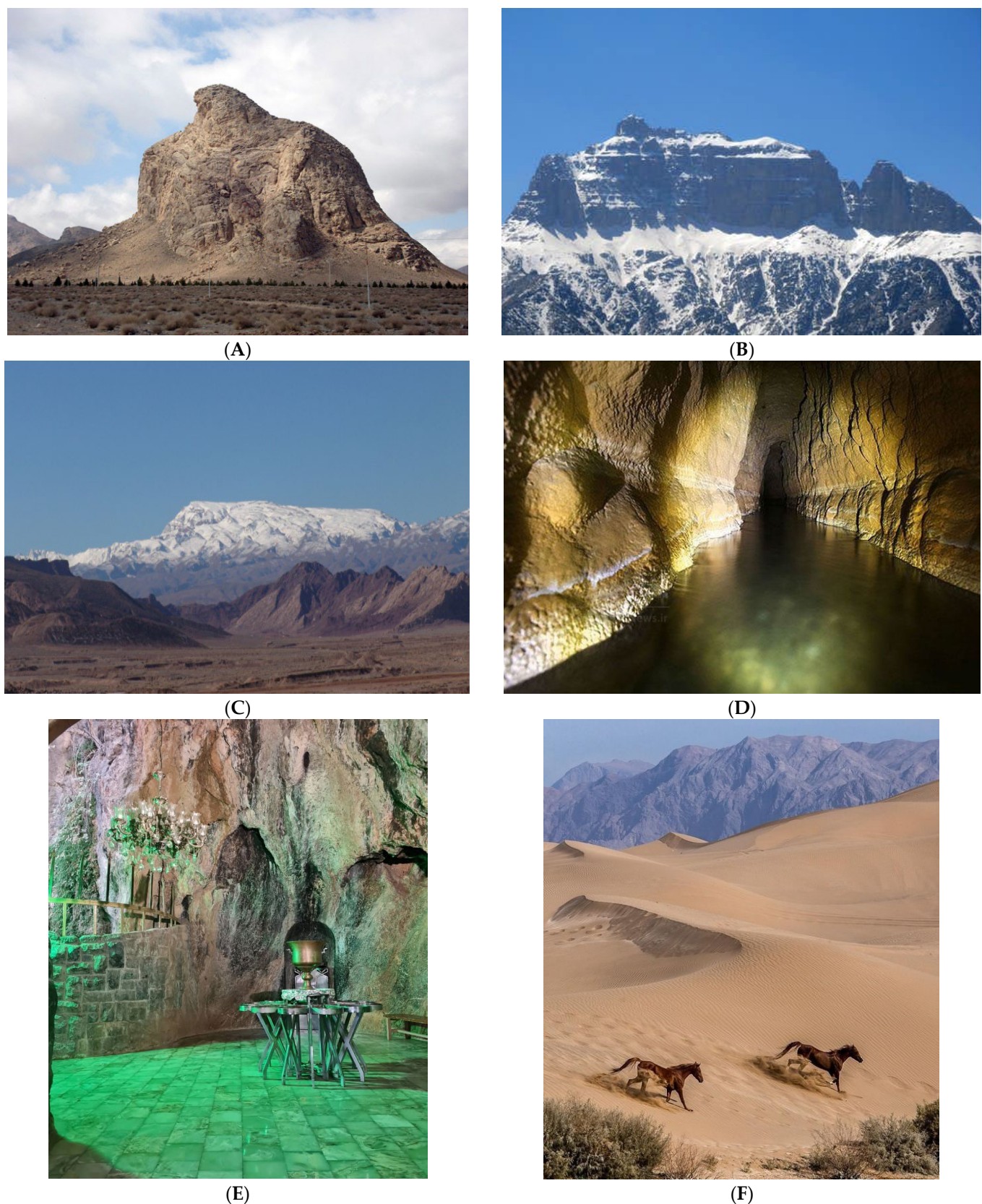

**Figure 4.** (**A**) The inselberg called "Eagle Mountain"; (**B**) Tejerjan Snow House; (**C**) Shirkooh Heights; (**D**) Aqueduct (Qanat); (**E**) Chak Chak (Photo source: S. Zareh®); (**F**) Siahkooh (Black Mountain, Desert, and Protected Area) (Photo source: J. Gorji®).

### 3.2.1. Eagle Mountain

Eagle Mountain, at a height of 2018 m, is located near the village of Islamieh along the Yazd-Shiraz route. Its unique appearance and beautiful views are among its tourist attractions. The rock material of this mountain is calcareous, and it is important from an educational and scientific point of view to understand the mechanism of differential erosion and weathering. According to the theory of experts and professors of geomorphology, the formation of this piece of limestone goes back to the beginning of the second geological period [80].

### 3.2.2. Barfkhane Tezerjan (Tezerjan Snow House)

Barfkhane is a Persian word that means the house of snow. The Tezerjan Snow House is located 18 km southwest of Taft city and 4.5 km southwest of Tezerjan village. The snow house in Tezerjan is a peak covered in glaciers and mountain glaciers that is popular for mountaineering and rock climbing. The effects of these glacial circuses, or snow houses, are significant in environmental and paleoclimate studies (adapted from [80]).

### 3.2.3. Shirkooh Heights (Moraines and Glacier Valley)

Shirkooh heights is located in the centre of Iran, 40 km south-southwest of Yazd province. With a height of 4075 m, this mountain is the highest in the region [81]. Monzogranite units comprise the largest batholith of Shirkooh Mountain [82]. Among Shirkooh's geomorphographic attractions, there are the moraines, glaciers, glacial circuses, and beautiful glacial valleys. The glacial moraines, scattered moraines (1800 m above sea level), circuses, and glacial valleys present the Shirkooh region as the most attractive natural museum in the world [57].

### 3.2.4. Aqueduct (Qanat)

Qanat (Kariz) is one of the ancient innovations devised by Iranians for extracting from aquifer tables and transporting fresh water to settlements [6]. The presence of aqueducts in the Yazd-Ardakan Plain is, for the area, more a form of anthropogenic geomorphology and less a natural attraction. These aqueducts are visible throughout the Yazd plain of Ardakan. This is about combining local geotourism and ecotourism offerings, leveraging the influence, and adapting Yazd's architectural heritage to capitalize on the water resource for both residents and tourists [83].

### 3.2.5. Chak Chak

There are several Zoroastrian shrines in Yazd, but the most famous is Chak Chak. Thousands of domestic and international tourists flock to the shrines each year particularly from 13 to 17 June. Chak Chak, with its stalactite and stalagmites and their cauliflower crystals, is another of this province's attractive natural landscapes [57].

### 3.2.6. Siahkooh (Black Mountain, Desert, and Protected Area)

The walls of a magnificent old mansion are visible from afar on the northern slopes of the Siahkooh (1865 m), the region's highest peak. Constructed of large rocks in the manner of Persepolis's famous structures, it was named the mother of Iranian caravanserais [84]. These traditional features of rural architecture can also be found in some man-made structures such as farms and recreational villages, eco-camps with facilities for ecotourism activities in the arid area with dune accumulations at the edge of Black Mountain (Figure 3). In the same mountainous environment, in arid or humid ecosystems, the positive experiences are enhanced by the exercise of discovery and frontalization with perfectly ecologically integrated biogeographic elements, some of them rare (e.g., *Caryophyllaceae* family, a type of plant from the clove family) in Shirkooh Heights [85], *Felis caracal* (a type of desert cat), Persian cheetah (*Acinonyx jubatus venaticus*), houbara bustard (*Chlamydotis undulata*, a special species of bird) etc.

### 4. Pralong Method

The present study involves a theoretical part, but it also has an applied aim, using a descriptive analytical method and field research (direct observation, field reports). The statistical units of this study are the geomorphosites/geosites of the Yazd-Ardakan Plain in Yazd Province. The tourist value of a site is calculated using this method as the average of its scenic, scientific, cultural, and economic values [86]. A specific criterion was used to determine the value of each indicator in this case.

There is no reason to weight one item more over another when determining the theoretical potential of an identified item to spur geomorphological landform tourism, as there is no compelling reason to believe that one index is more telling or significant than the other (although the "Cultural Value" parameter receives some extra weighting due to the potential of a literary biography of the geomorphological landmark [66]).

Accordingly, tourism-related potentials of a landform can be expressed through the means of the four indexes as follows.

The scientific value of a geomorphological landform is calculated based on criteria such as rarity, educational status, and paleo-geographical and biological value.

It can be calculated according to the following equation and the rates presented in Table 1. Scientific value = $(V1 + V2 + 0.5 \times V3 + 0.5 \times V4 + V5 + V6)/5$. The scenic value of a geomorphological landform depends on its inherent scenic aspects and can be calculated according to the following equation as well as by the rates in Table 1. Scenic value = $(V1 + V2 + V3 + V4 + V5)/5$. When assessing the validity of historical–cultural elements on the depth of artistic ability/expression, the emphasis is on artistic ability and cultural mores, prevalent in geomorphologic places. In this formula, the weight of paragraph 2 (V2: iconographic representations in Table 1 related by cultural value) is calculated twice, as this paragraph may also include a literary biography, usually associated with the iconography of 1. Points are calculated in accordance with Table 1. Cultural value = $(V1 + 2 \times V2 + V3 + V4 + V5)/6$. When assessing socio-economic empowerment, the emphasis is on useable features and the entrepreneurship of the item in the field of landform tourism. Points are calculated based on Table 1 as well. Economic Value = $(V1 + V2 + V3 + V4 + V5)/5$.

After scoring a given landform, the assessment of the degree of exploitation for the geomorphological landforms was examined. This assessment consisted of two components and was homological to the tourism capability assessment, with criteria and scoring scales specified for each of the components.

Accordingly, the stated degree of exploitation (coordinate X) and quality (coordinate Y) of exploitation were both given; therefore, the coordinates were developed according to the following equation: exploitation value = degree of exploitation value, modality of exploitation value, where the degree of the exploitation value represents the spatial and temporal use of the geomorphological landform and can be drawn from the following equation, with its rating being calculated according to Table 1. The degree of exploitation value: $(V1 + V2 + V3 + V4)/4$.

Additionally, the quality of the exploitation value was calculated on the basis of using four geomorphologic landform tourism score criteria with its provisions calculated according to Table 1. The modality of the exploitation value = $(V1 + V2 + V3 + V4)/4$.

Following identifying the geomorphological features that were introduced as tourist attraction capabilities in the Yazd-Ardakan plain and using the Pralong method to analyse the factors affecting the development of tourism in the region, a table of areas with the ability to be geomorphosite was prepared and scored, and finally the amount and percentage of each place was specified.

Using this method, the desired geotourism and ecotourism areas were identified and placed in the form of a questionnaire at the disposal of experts. After presenting the polls, a final table and scoring system for the desired sites were designed.

**Table 1.** The evaluation of the four geomorphosite/geosite criteria and the value of tourism productivity, based on the Pralong method.

| Value | | Criterion-Score | 0 | 0.25 | 0.5 | 0.75 | 1 |
|---|---|---|---|---|---|---|---|
| **Scientific** | V1 | Paleo-geographical interest | - | Low | Modest | High | Very high |
| | V2 | Representativeness | Zero | Low | Modest | High | Very high |
| | V3 | Area (plottage) [%] | - | Less than 25 | 25–50 | 50–90 | More than 90 |
| | V4 | Rarity (rareness) | More than 7 | 5–6 | 3–4 | 1–2 | Unique |
| | V5 | Integrity | Destroyed | Strongly deteriorated | Moderately deteriorated | Weakly deteriorated | Intact |
| | V6 | Ecological interest | Zero | Low | Modest | High | Very high |
| **Scenic** | V1 | Number of view points | - | 1 | 2–3 | 4–5 | More than 6 |
| | V2 | Average distance to viewpoints [m] | - | Less than 50 | 50–200 | 200–500 | More than 500 |
| | V3 | Surface | - | Small | Modest | Large | Very large |
| | V4 | Elevation | Zero | Low | Modest | High | Very high |
| | V5 | Color contrasts with site surroundings | Identical colors | - | Different colors | - | Opposite colors |
| **Cultural** | V1 | Cultural and historical customs | Without link | Weakly linked | Moderately linked | Strongly linked | Initiatory of custom(s) |
| | V2 | Iconographic representations | Never represented | Represented 1 and 5 | Represented 6 and 20 | Represented 21 and 50 | Represented 50 or more |
| | V3 | Historical and archaeological relevance | No vestige or building | Weak relevance | Modest relevance | High relevance | Very high relevance |
| | V4 | Religious and metaphysical relevance | No relevance | Weak relevance | Modest relevance | High relevance | Very high relevance |
| | V5 | Art and cultural event | Never | - | Occasionally | - | At least once a year |
| **Economic** | V1 | Accessibility | More than 1 km of track | Less than 1 km of track | By a local road | By a road of regional importance | By a road of national importance |
| | V2 | Natural risks | Uncontrollable | Not controlled | Partially controlled | Controlled residual | No risk |
| | V3 | Annual number of visitors in the region | Less than 10.000 | 10–100.000 | 0.1–0.5 million | 0.5–1 million | More than 1 million |
| | V4 | Official level of protection | Complete | Limiting | - | Not limiting | No protection |
| | V5 | Attraction | - | Local | Regional | National | International |
| **Degree of exploitation** | V1 | Used surface [ha] | Zero or ex situ | Less than 1 | 1–5 | 5–10 | More than 10 |
| | V2 | Number of infrastructure | Zero or ex situ | 1 | 2–5 | 6–10 | More than 10 |
| | V3 | Seasonal occupancy [day] | - | 1–90 (1 season) | 91–180 (2 seasons) | 181–270 (3 seasons) | 271–360 (4 seasons) |
| | V4 | Daily occupancy [hour] | - | Less than 3 h | 3–6 | 6–9 | More than 9 h |
| **Quality of exploitation** | V1 | Use of the scenic value | No advertising-optimization | 1 support and 1 product | 1 support and some products | Some means of support and 1 product | Some means of support and products |
| | V2 | Use of the scientific value | No didactic optimization | 1 support and 1 product | 1 support and several products | Several means of support and 1 product | Several means of support and products |
| | V3 | Use of the cultural value | No didactic optimization | 1 support and 1 product | 1 support and several products | Several means of support and 1 product | Several means of support and products |
| | V4 | Use of the economic value [person] | No visitor | Less than 5000 | 5000–20,000 | 20,000–100,000 | More than 100,000 |

## 5. Results and Discussion

The Pralong model was used to compare the credibility, value, and tourism capabilities of selected geomorphosites within the study area. Calculating the tourism value and productivity of the region's landforms enables a comprehensive understanding of the region's landforms' capabilities. Additionally, taking into account the region's other potential and tourism potential, such as natural, human, historical, and ecological attractions, among others, it has provided solutions that are appropriate to the region's ability to attract tourists.

After conducting detailed studies in the region and using the opinion of experts, the table of tourism and interest values associated with geomorphological landscapes was prepared and completed, and necessary evaluations were made.

The present study employed the Pralong method, as did other studies [37,62,66,67], and its results indicated that using the Pralong method to study geotourism and regional

tourism is appropriate and has produced results for a variety of tourism sites in the study area. Geomorphosites can also be prioritized using the Pralong method, such as similar works that were conducted in different regions of the world, where the relationship between geomorphosites and landscapes matters [87]. The significance of tourism in the geomorphological sites of the plains of this province and about attracting tourists can be found in Omidvar's article [57]. Table 2 shows the overall results of this study after collecting the questionnaire and expert opinions and using the Pralong method. According to the obtained results, the geomorphosites Chak Chak, with an average tourism grade of 0.62, and Shirkooh Heights (moraines and glacial valleys), with a score of 0.61, were ranked first and second, respectively, in terms of tourism grade. Additionally, other geomorphosites were nearly identical in terms of tourism and were generally ranked lower, with Siahkooh Desert having the lowest numerical value of the standard value of tourism at 0.51. In terms of productivity value, Chak Chak with an average score of 0.52 and Shirkooh Heights (moraine and glacial valleys) with a score of 0.51 were in the first and second ranks, although other regions with the same average productivity value were in the next ranks. Meanwhile, Mountain Eagle received the lowest productivity value score of 0.39.

**Table 2.** The evaluation of tourism and interest values on existing iconic geomorphological landforms in the region.

| Standard-Landform | Shirkooh Heights (Moraine and Glacier Valley) | Black Mountain (Siahkooh, Desert and Protected Area) | Chak Chak | Aqueduct/ Qanat | Tezerjan Snow House | Mountain Eagle |
|---|---|---|---|---|---|---|
| Appearance aesthetic value | 0.82 | 0.60 | 0.49 | 0.50 | 0.65 | 0.53 |
| Scientific value | 0.72 | 0.67 | 0.60 | 0.58 | 0.61 | 0.63 |
| Historical-cultural value | 0.28 | 0.26 | 0.77 | 0.55 | 0.32 | 0.32 |
| Socio-economic value | 0.61 | 0.51 | 0.61 | 0.59 | 0.5 | 0.61 |
| Average tourism grade | 0.61 | 0.52 | 0.62 | 0.55 | 0.62 | 0.52 |
| The value of productivity | 0.64 | 0.53 | 0.56 | 0.49 | 0.5 | 0.44 |
| Product quality value | 0.4 | 0.36 | 0.49 | 0.45 | 0.33 | 0.34 |
| Average productivity value | 0.51 | 0.44 | 0.52 | 0.47 | 0.41 | 0.39 |
| Qualitative scale | Medium | Medium | Medium | Medium | Medium | Medium |

Finally, it is necessary to treat all these numerical results within an interdisciplinary approach to obtain a clear understanding of the factors that shape the promotion opportunities available to economic and social actors interested in these geotourism and ecotourism resources (Tables 3 and 4). The promotion is based on the expectations and needs of tourists [88], which are satisfied when contact with a vivid image of a tourist attraction generated in their consciousness the character of uniqueness, distinction, motivation, and, implicitly, affective reaction experiences [89,90]. Each of the tourist resources analysed in this paper are, beyond the obvious geological and geomorphological register, landscapes reunited in a landscapes complex [91] (p. 139). The production of "beauty" (especially particular), of aesthetics [92–95], or of the "monotonous, banal, repulsive" of the landscape depends on the capacity of the individual/observer and on the exercise of one's mind to capture the abstract message of the formal properties of the landscapes of a territory. Man as a viewer and as an observer [96] and the tourist as an observer and consumer of landscapes must be allowed to roam the fertile ground of the discovery of beauty and aesthetics wherever they are and however they manifest themselves.

These landscapes [91] (pp. 176–178) [9,96] are treated as tourist geo-destinations by eco- and geotourists [97,98] by revealing the iconic meaning of their destination capacity [11,99]. All these factors contribute to the attractiveness and competitiveness of these landscapes as parts of a whole or as a whole, which are reinforced by image elements [76–79] manifested through aesthetic and cultural values [39] as well as emotional reactions.

**Table 3.** The organisation and personality of landscapes expressed through visual composition.

| Geosites/ Geomorphosites | The Main Geotourism and Ecotourism Attractions | The Character and Visual Composition of the Landscapes |
|---|---|---|
| **Mountain Eagle** | *Geomorphologically relevant geosites (geomorphosites):* Association of rounded or sharp mountain interfluves and erosion witnesses suggestive of eagle shape | Generous opening angles of 90–110°; irregular volumes (eagle-like block) and elongated prismatic horizontally (the rest of the mountain assembly), slightly; zigzagged lines of force; structuring axes alternating between the horizontal register (the main interfluves) and the vertical one (secondary interfluves, valleys, and sloping rock layers); fine coarse texture and neutral beige chromatic register both for erosion witnesses and for the rest of the mountain |
| **Barfkhane Tezerjan (Tezerjan snow house)** | *Geosites of mixed geomorphological and hydrological relevance:* Association between snow, glaciers, and the result of snow-glacial modeling (valleys, peaks, stone walls, scree, cavities, and deep cracks in the rock) | Pronounced three-dimensional character, with irregular, rounded, or angular solid volumes, but also for flattening (snow volumes on plateaus, suspended circuses or avalanche color); zigzag-elongated lines of force (suspended synclines); 2–3 main landscape plans; simplified chromatics in shades of gray and white |
| **Qanat (Aqueduct)** | *Mixed geomorphological and hydrological geosites:* Underground development reflecting particular hydrogeological conditions materialized in inclined aqueduct/Qanat, arranged underground on two levels/floors; soft rock in which the aqueduct was dug; water differs in hardness and temperature (on the upper floor it is sweet, light and cold; on the lower floor it is warmer and heavier); vents and water recovery; structures and material for lining the inlets/outlets of the aqueduct | Shapes and volumes close to semi-cylindrical or semi-ellipsoidal, elongated and sinusoidal, tiered; slightly fluid horizontal plane that gently widens towards the exit (groundwater flow); narrow opening angle and aisle effect; coarse texture with small cavities (rock walls); and distinctly smooth with regular geometric pattern (brick lining on the walls at the outlet) |
| **Chak Chak** | *Geosites highlighting geomorphosites with religious and historical significance:* The geomorphological relevance of the site marked by the cavity that houses the Zoroastrian altar and the steep, strongly altered mountain slope and the access alley to the altar with a serpentine route; the altar and the objects of worship; the spring considered holy, fires kept forever burning; the complex of buildings that serves the religious ensemble, including pilgrims; the steep, strongly altered mountain slope; and the access alley to the altar with a serpentine route | For the interior of the altar, there is a small opening angle (below 90°) and limited viewing axis (10–15 m); blocking effect (the gaze is limited due to cavity walls); for the exterior of the altar, there is a maximum opening angle (120°); viewing axis of the order of 2–6 km; maximum panning effect; depression/depth angle of 30–45° (top view from terraces-built platforms); elevation angle of 35–55° (view from the base of the slope or from the slope); coarse texture; dominant beige color palette |
| **Siahkooh Black Mountain (Desert and Protected Area)** | *Geomorphologically relevant geosites (geomorphosites):* The black volcanic mountain; swamp with formations of adapted wet vegetation (only in the rainy season); desert sand dunes; wildlife perfectly adapted to the environment, including endemics (small-, small-medium-, and medium-sized mammals, wet or semi-arid environment birds, reptiles) | The association of spacious open volume (the desert and the part that temporarily hosts the swamp) with a pronounced volume articulated vertically (mountain ridge and piedmont); viewing axes (4–12 km) and very generous opening angles (around 120°); black associative chromatic register (mountain)-scarlet (piedmont); 1–2 main landscape plans (for dunes and swamp) and 2–3 main plans for mountains and piedmont; island position for volcanic mountains; coarse texture for mountains (scree trains) and fine for other ecosystems |
| **Shirkooh Heights (Moraine and Glacier Valley)** | *Mixed geomorphological, botanical and faunal geosites:* Sharp main interfluves, rounded secondary interfluves, alpine plateaus, glacial circuses, glacial valleys, moraines, stone mattresses, scree trains, and erosion witnesses; isolated rocks; suspended synclines; micro terraces; bushes and tufts of shrubs and plants with multicolored and rare flowers all present in the alpine floor; snow or ice blanket; slope springs; valleys and canyons, stormy streams, and small waterfalls in spring-summer | Mountainous mass developed similar to a horseshoe horizontally; grandiose volume and impetuous vertical development; lines of force with rare changes in their orientation; dense structuring inclined lines in almost vertical slopes (little packages of geological strata); (opening angles to 120° (only on valleys and canyons the value is between 60–70°); viewing axes between 6 km (E-V) and 10 km (SW-NE); impressive number of viewpoints; mostly coarse texture moderate, simple chromatic and in antithesis dark white-gray mountainous mass developed similar to a horseshoe horizontally; grandiose volume and impetuous vertical development; lines of force with rare changes in their orientation; dense structuring inclined lines in almost vertical slopes (little packages of geological strata); (opening angles to 120° (only on valleys and canyons the value is between 60–70°); viewing axes between 6 km (E-V) and 10 km (SW-NE); impressive number of viewpoints; mostly coarse texture moderate, simple chromatic, and in antithesis dark white-gray |

Tourists who visit the landscapes of these geosites are interested in interacting and socializing with them so that the satisfaction of geotourism and ecotourism is maximized. In an effort to put in order the situation of correct understanding of the identity character of the landscapes in the area, to achieve geoeducation by refining the interpretation and thinking of visitors [99,100], to contribute to geoconservation and for the best visitor

experience [44,101,102], and to the training of guides, we advance the landscape treatment that justifies the results of applying the method of geomorphosite evaluation (Tables 3 and 4).

**Table 4.** The identity of the landscapes in the area defined by the emotions and reactions developed by tourists, contributing to the promotion and practice of geotourism and ecotourism.

| Geosites/ Geomorphosites | Destination Capacity through Aesthetic Values and Emotional Reactions | Who Is It For? | Forms of Geotourism and Ecotourism Practiced or That Can Be Practiced |
|---|---|---|---|
| **Mountain Eagle** | Strong individual experiences and satisfaction developed on account of the diversity, the scope of the relief forms, and the majestic character of the landform (inselberg) | Dominant to very young tourists and young adults, energetic, active, and eager to socialize immediately with the mountain | Scientific, observational, and discovery; contemplative tourism; camping; hiking and mountaineering; climbing; tourist orientation; trekking |
| **Barfkhane Tezerjan (Tezerjan snow house)** | Strength and energy in a comforting combination, an accentuated spirit of freedom of sight developed at the expense of the organic, harmonious integration of snow and rocks | Dominant to young tourists, well equipped and physically prepared, with sports touches, professional or amateur, energetic, active, and eager to front and socialize immediately with the mountain | Scientific, observational and discovery; contemplative tourism; camping; hiking and mountaineering; climbing; trekking |
| **Qanat/ Aqueduct** | Unity and contrast of the masses; chromatic of neutral tones for the underground walls, not to mention in their case of repulsive character; attachment to the underground nature and to the result of the competence of local builders and simple freshwater management; security despite the underground placement of the ensemble | Tourists of all ages eager for guidance to experience live knowledge of the means, techniques, and results of underground aqueduct planning; to try water through individual tasting experiences; sensory education by identifying tourists with a comforting underground environment compared to the slightly restrictive one on the surface | Tourism through water-related activities; scientific, observational, and discovery; leisure tourism |
| **Chak Chak** | The evocation of the atmosphere of harmony between the petrographic nature of the cave and the essence of Zoroastrian spirituality; identification through prayer of the tourist/pilgrim with the deities; the representation through the sober colors, the objects of worship and the atmosphere of the divine absolute; balance and peace through faith | For pilgrims and tourists interested in religion, history, the philosophy of religion, and the culture of the Middle East, but also for landscapes that open from natural plateaus and built platforms, especially young people and adults but less elderly (the last part of the route has a high level of difficulty) | Vent tourism; pilgrimage and cultural-religious tourism; historical tourism; food tourism; tourism for panoramas and landscapes; treasure hunt |
| **Siahkooh Black Mountain (Desert and Protected Area)** | Unity and contrast of the masses; pleasant and tonic ensemble; harmonious integration of water volume, aquatic vegetation, sand dunes, foothills, and mountains; pleasant dispute between volumes and lines; dispute of cold and warm tones, not to mention the repulsive character; daydreaming mood; proving the idea of aspiration to infinity; search for the "self" | Tourists educated in the spirit of nature interested in discovering the morphological details of the abiotic part (deposits of altered or unaltered volcanic rock and desert sand) and biotic part (local wildlife) | Camel and horse riding, paragliding, safari, motorcycling, skydiving, quad biking, camping; sand-boarding; hiking and mountaineering; bird watching; sky watching (with and without telescope) |
| **Shirkooh Heights (Moraine and Glacier Valley)** | Pleasant dispute between volumes and lines; evocative plasticity by resembling a lion sleeping on its paws, generated by the association between the mountain mass, the piedmont, and the impressive glacial valley; attachment to mountain nature; security despite the construction of the high voltage key assembly | Especially young tourists, amateurs, and professionals, of good and very good socio-professional condition, with ecological awareness and maximum openness to discovering the local nature, who want to discover through physical effort coupled with rest sessions the benefits of grandiose geomorphic-biogeographic organisation of an eminently mountainous space | Hiking to geological wonders; trekking and mountaineering among a sanctuary of wildlife; camping; rock climbing; paragliding and hang-gliding (possible launches from the main ridge to the east to the villages of the Najib Valley); adventure skiing; rural tourism (Deh Bala, Tezerjan); awesome landscape photography sessions |

The methods applied in this study led to concrete results concerning the geographical unit of interest. It presents geosites whose evaluation revealed values of potential for tourism, be it geotourism or ecotourism, which can be accredited, and the results only accredit these landforms at the national, not international, level. This study was an opportunity to question the change of the paradigm "sensational and exceptional" at any cost in tourism. We can and must talk about and promote a new culture of supply, choice, and taste orientation. Scientists must contribute to education and become more than just exponents of applications in which preciousness and academism abound that are difficult for laymen to understand, here calling them tourists. Thus, after all, we should be the first to overcome this phase of "typological discrimination". The result would be that even geosites of average value, and even below average values, will arouse curiosity, become of interest, and gain relevance in the act of choice by visitors.

*Proposals for Thematic Activities and Infrastructure for Geotourism and Ecotourism*

Here are some suggestions on how to look or obtain an appointment for geotourism and ecotourism in the future:

- Carry out feasibility studies towards the valorisation of landscapes through geotourism and ecotourism in different sites, for the educational role in proactive and participatory planning and for the economic and cultural development of indigenous people in the region (as is done in the parks of the Australian Alps [103], and for China [104]);
- Prepare support and training packages, introducing geosites (and their landscapes), and creating awareness among people to use geotourism and ecotourism areas instead of artificial tourism areas;
- Build attractive geotourism and ecotourism thematic trails/geoitineraries between Shirkooh and Siahkooh, with tourist paths and shelters, an information centre, information and interpretation panels with text levels and descriptive graphics, orientation signs, travelling exhibition, and natural and built platforms/viewpoints for viewing landscapes (example being the Italian experience of geoitineraries in the southern Apennine Mountains [105–107]; the German experience in UGGp Swabian Alb Geopark [108]; or the Austrian experience in the Alps [45]);
- Preserve existing eco-camps and develop a network with related facilities (Figures 4 and 5);
- Construct a tourist camping area in Shirkooh, including a creative camp for art with themes related to landscapes marked by local geoheritage and ecoheritage, through natural and cultural heritage, wildlife habitats, and recreation;
- Provide parking spaces that allow access and safe access to the most popular landscapes, both by transport for large groups and individually;
- Conduct landscape science courses for students and guides in different fields and categories, especially in geography, environmental science, tourism planning, and tourism management, practicing through an educational tour in Yazd-Ardakan Plain;
- Implement the website by the partners managing and exploiting the geosites' landscapes, including the promotion of the use of catering, local products, and cultural-religious activities.

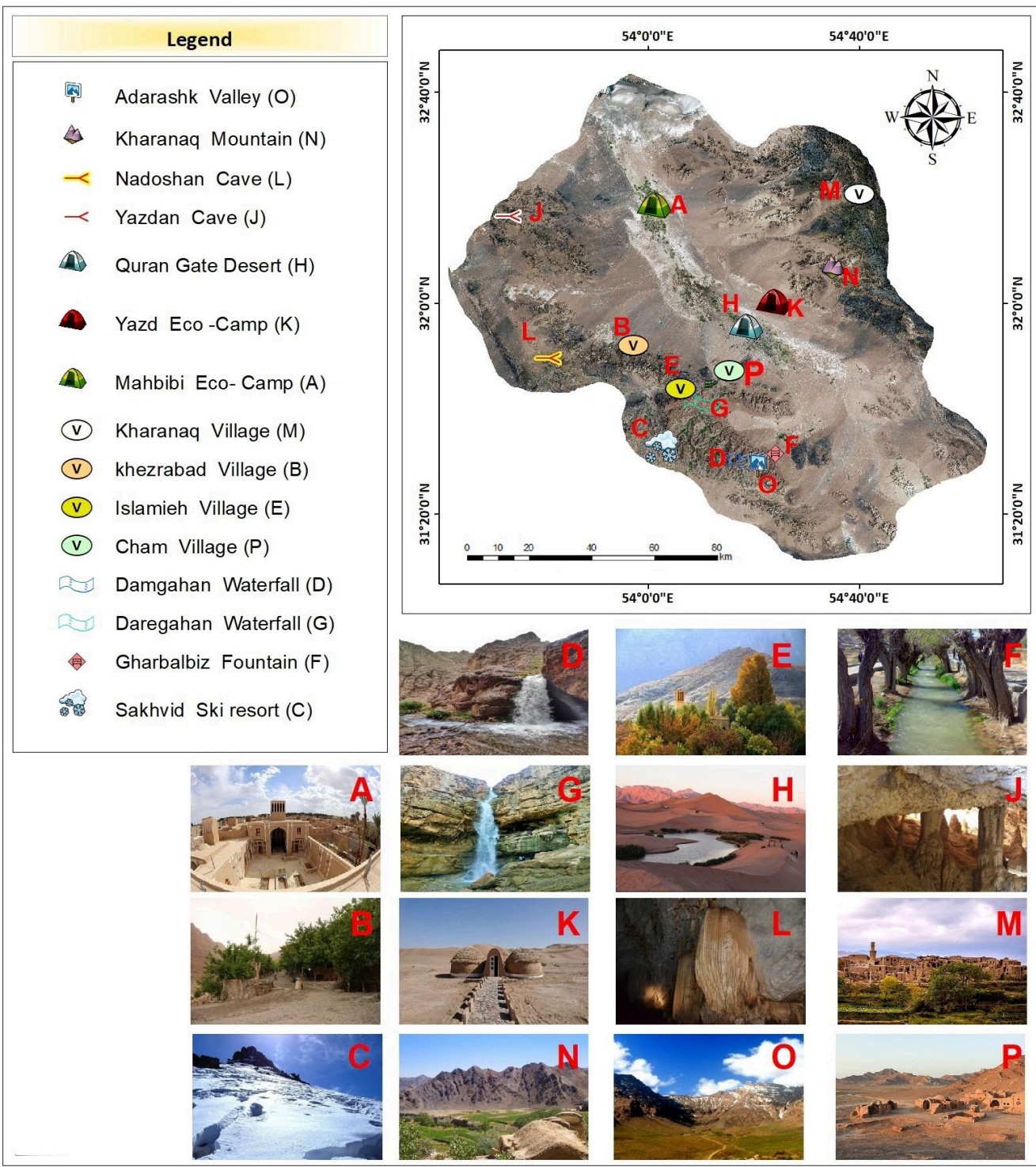

**Figure 5.** The existing network of eco-camps on the Yazd-Ardakan Plain, with specific infrastructure and a great variety of landscapes supported by a variety of geosites.

For the smooth running of these experiential programs, geotourists and ecotourists will benefit from low and high infrastructure for local geo- and ecotourism, as well as adapted residential facilities [109] (Figures 5 and 6). These programs and service infrastructure will align with efforts to develop and protect local geosites and landscapes, including academic and professional initiatives such as the comprehensive study of Shirkooh to become a geopark, with the mountain unit being placed in the final stage of approval by UNESCO [110]. This measure, made concrete and strongly popularized both online and

through appropriate marketing measures, by enriching the portfolio of local travel agencies and large international tour operators, will increase interest particularly for foreign tourists. The further development of the transport infrastructure to the attractions, the infrastructure for sightseeing and accommodation, and the level of difficulty of the sightseeing-discovery routes, which are largely open to all tourists, will be able to support geotourism and ecotourism at responsible and sustainable levels.

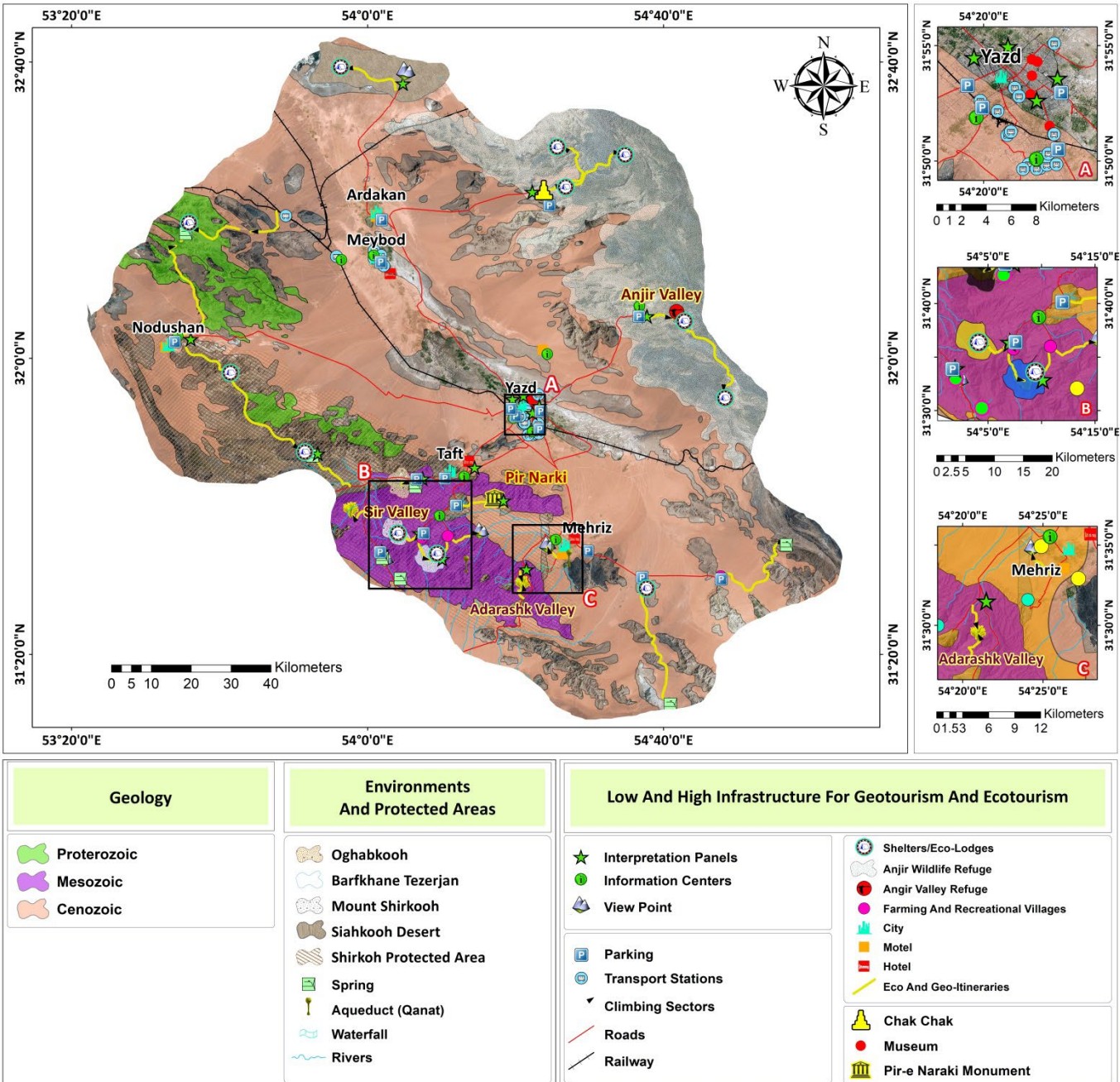

**Figure 6.** Low and high infrastructure, as well as adapted residential facilities, for the practice of local geotourism and ecotourism.

## 6. Conclusions

In comparison to other regions in the world [101–118], Yazd province's geological diversity, topographical conditions, biodiversity elements, larger climate changes, and landscapes make it a natural and iconic destination for Iranian and international ecotourists [57].

At the operational level, with a few exceptions (Chak Chak, the Qanat/Aqueduct), the working hypothesis concerning the conditioning relationship between the variables petrographic nature of the substratum, geosite value, and landscape expression strength was always demonstrated. Thus, the results showed a correlation between hard rocks, steep and imposing landforms, advanced modelling phenomena, and the quality of geosites and landscapes for geotourism and ecotourism corresponding to the interfluves, slopes, moraines, and glaciers of Shirkooh, Eagle Mountain, Tezerjan Snow House, and Siahkooh. These are the variables of geomorphosites and landscapes where force and tension are exciting, positive affective-emotional experiences released by the harshness of sublime landscapes, solidity, the vastness of mountain masses, the unique structural conformation of glacial valleys, steep and very steep slopes, glacial deposits, and the organisation of water in a solid state (snow blanket or glaciers).

A special situation is that of Chak Chak and Qanat, where the high value of the potential for tourism of the relief is not explained by the deterministic effect of the rock, geosite, and landscape relationship. Here, the softer rocks create attractions for geotourism and ecotourism through accessibility, simplicity, fascination, impact, and captivating experience due to the interference between the purely natural part and the part marked as constructed nature.

According to the results, in general, the average size of all geosites with their landscapes in terms of tourism and productivity shows the region's ability to attract a future for tourists. Another point to consider in this regard is the province's high diversity of tourism in all biophysical and geological aspects, including landscape units [43,119], which causes tourists to disperse across all these areas. The landscapes of the mountain units on the SW and NE sides of the Yazd-Ardakan Plain are of added geotouristic and ecotouristic attraction.

All this evidence leads to the need to connect these geosites with landscapes, where geotourism and ecotourism are currently practised or that have good opportunities for geotourism and ecotourism, to a more than local network, which can enhance tourist and economic development and knowledge exchange [120].

**Author Contributions:** Conceptualization: S.R.K., I.D. and S.A.A.; methodology: S.R.K., I.D. and S.A.A.; software: S.R.K., I.D. and S.A.A.; validation: I.D. and S.A.A.; formal analysis: S.R.K., I.D. and S.A.A.; investigation: S.R.K., I.D. and S.A.A.; resources: S.R.K., I.D. and S.A.A.; data curation: I.D. and S.A.A.; writing—original draft preparation: S.R.K.; writing—review and editing: I.D. and S.A.A.; visualization: S.A.A.; supervision: I.D.; project administration: S.R.K. and S.A.A.; funding acquisition: I.D. All authors have read and agreed to the published version of the manuscript.

**Funding:** This research was funded by the University of Oradea. The field research activities were carried out through the Erasmus+ program no. 2018-1-RO01-KA107-04_2018, Key Action 1.

**Data Availability Statement:** Data are available in a publicly accessible repository. The data presented in this study are openly available in the repository LAND, Special Issue "Landscape Heritage: Geomorphology, Geoheritage and Geoparks".

**Acknowledgments:** The authors are grateful and would like to thank the academic editors and to anonymous reviewers for their valuable comments on the manuscript.

**Conflicts of Interest:** The authors declare that they have no conflict of interests.

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
