# Peer review of "Landscapes of the Yazd-Ardakan Plain (Iran) and the Assessment of Geotourism—Contribution to the Promotion and Practice of Geotourism and Ecotourism"

_land, doi:10.3390/land12040858_

Round 1
Reviewer 1 Report (Previous Reviewer 1)
Dear authors! The article is informative and well-written. I hope the methods will be useful for other research. Thanks to the authors for their work. I wish you success in your creative work.
Author Response
Dear Prof.,
We thank you for your kind words, sincere encouragement and all your effort in reading and reviewing our manuscript.
Wish you all the best.
Reviewer 2 Report (Previous Reviewer 3)
This reviewer stands by the general comments made in the first revision of this article. This is not a research article, nor is it a review article in the strict sense of the term. There is no hypothesis to be contrasted, nor is a method proposed to identify and evaluate the tourist interest in geomorphological sites. The authors apply methods proposed by other researchers, but without critically evaluating their usefulness, successes, and failures. The territory where the work is applied is not excessively relevant internationally; in fact, most of the evaluated geomorphological sites have average ratings according to the applied method. Therefore, the work seems very relevant at the national level but not at the international level. The references used are adequate and up-to-date.
The use of the concept of ecotourism in the article is not sufficiently justified in the introduction or in the research framework. The methods adopted are to evaluate the interest in tourism and to analyze the landscapes visually, but no methods are applied to evaluate ecotourism. Also, this reviewer wonders if geotourism is not part of ecotourism. Therefore, it is proposed to eliminate the concept and the term ecotourism from the article.
The conclusions are a mixture of a summary of the article and personal reflections of the authors since they do not really validate any hypothesis since the article does not present any working hypothesis. It is a work based on the application of methods, not designed by the authors, to evaluate the geotourism potential of a specific territory in Iran. It should be noted that the authors already published an article on the geotourism interest of Yazd province in 2010.
The authors do not yet cite the main article where Pralong states the method used here. The Pralong article to be included is: https://doi.org/10.4000/geomorphologie.350
This reviewer acknowledges the efforts made by the authors to improve all figures in accordance with the recommendations offered in the first review. In relation to figure 1, the authors should unify the representation of the altitudes without distinguishing between Shirkooh and Ardakan, and Anjir.
When developing the graphic scale of the figures contained in the maps, the authors refer to them sometimes in miles and sometimes in kilometers. Once they decide whether to use miles or kilometers, they should use the internationally accepted abbreviation mi or km on the graphic scale and not miles or kilometers.
Author Response
Dear Prof.,
Thank you for your patience and effort in reading and reviewing our manuscript. We assure you of our appreciation for the advice and recommendations made to improve the quality of the manuscript.
We wish you all the best.
Comments and Suggestions for Authors
This reviewer stands by the general comments made in the first revision of this article. This is not a research article, nor is it a review article in the strict sense of the term. There is no hypothesis to be contrasted, nor is a method proposed to identify and evaluate the tourist interest in geomorphological sites. The authors apply methods proposed by other researchers, but without critically evaluating their usefulness, successes, and failures. The territory where the work is applied is not excessively relevant internationally; in fact, most of the evaluated geomorphological sites have average ratings according to the applied method. Therefore, the work seems very relevant at the national level but not at the international level. The references used are adequate and up-to-date.
Answer. The Pralong method, although not applied today by all researchers assessing the tourism potential of geosites, is still of wide interest. This is explained by the fact that the method uses the six indicators, the experience and results not only of the method's proponent, but also the experience and results of other studies published before Pralong. In addition, the benchmark indicators taken into consideration in this study fit very well with our intentions of multi- and interdisciplinary analysis and thematics. In other words, the first four indicators gather the convergent interest for the component part and the socio-economic-cultural relevance of geosites, and the last two indicators outline the openness for valorisation of the same geosites. Perhaps only the economic indicator carries some critical discussion, in the sense that subjectivity factors into what is meant by the economic value of geosites. It may be irrational rapid economic exploitation that carries risks for the conservation of geosites, but we can also think of long-term economic exploitation, where financial benefits come in tandem with care for the existence and condition of the same geosites.
The methods proposed by Jakel and Bell and used in this study incorporate a certain degree of subjectivism, but one that is unanimously accepted in landscape science. Although these methods emphasize the design of form and function of landscapes that relate to geosites, they are the ones that leave behind the sometimes exaggerated academism of classic non-landscape methods and shift the analytical meaning to the narrative of elementary and practical understanding of landscapes by visitors and tourists.
Yes, you are correct in stating that the geographical unit under study has geosites/geomorphsites whose assessment shows average values and that the results only accredit geosites at national, not international level. Fortunately there are courageous initiatives materialized in studies that highlight average scores (Baboli Moakhar and Ramesht, 2018) for the tourism and educational potential of other landforms. In fact, on the occasion of this study we must question the paradigm shift of seeking the "sensational and exceptional" at all costs. We can and must talk about and promote a new culture of supply, choice and taste orientation. It is we scientists who must become more than just exponents of academism that is difficult to digest by laymen. Herein lies the overwhelming merit of landscape science which treats each landscape unit by virtue of postulate thinking, in our case geosites, as analogous yet unique (J.C. Wieber, 1984; I. Mac, 1990). So, after all, we should be the first to overcome this phase of "typological separatism and discrimination", so that even geosites of average value become relevant in the act of choice on the part of those concerned.
J.C. Wieber, 1984, Apréhension et compréhension des paysages. XXVe Congres International de Géographie, in ”La recherche géographique francaise-Structures, Themes et Perspectives”, Paris-Alpes.
- Mac, 1990, Geographical landscape. Content and scientific significance (in Romanian). Terra, Revistă a Societății de Geografie din România, Anul XXII (XLII), Nr. 1-4, București.
The use of the concept of ecotourism in the article is not sufficiently justified in the introduction or in the research framework. The methods adopted are to evaluate the interest in tourism and to analyze the landscapes visually, but no methods are applied to evaluate ecotourism. Also, this reviewer wonders if geotourism is not part of ecotourism. Therefore, it is proposed to eliminate the concept and the term ecotourism from the article.
Answer. We understand the concerns expressed and we respect your point of view. However, in this study we did not propose to apply an ecotourism evaluation method, but only for geotourism, which has already been done by applying the Pralong method. For an evaluation of ecotourism a different system would have been necessary, using the Delphi method (with many variables and indicators, only 110 indicators). Ecotourism is included in this study only because of the existence of activities of profile, the concretization of activities, the consequence of the expression of a potential, then what matters are the considerations related to promotion. The central pillar of the study is in fact the landscape. They move, they empower and around them gravitate geoturism and ecotourism, because what are landforms, geosites in the eyes and minds of some tourists, if not landscapes. Both forms of tourism are in the path of a growing trend of opinion that they are close and complementary thematically, but each has its own field of expression. Hence the natural question: which of the two forms of tourism is more comprehensive? We say ecotourism, but our study focuses on geotourism.
This is also the reason why we agree to change the title of the article, the new form being: Landscapes of the Yazd-Ardakan Plain (Iran) and the assessment of geotourism. Contribution to the promotion and practice of geotourism and ecotourism.
The conclusions are a mixture of a summary of the article and personal reflections of the authors since they do not really validate any hypothesis since the article does not present any working hypothesis. It is a work based on the application of methods, not designed by the authors, to evaluate the geotourism potential of a specific territory in Iran. It should be noted that the authors already published an article on the geotourism interest of Yazd province in 2010.
Answer. We have intervened in the Conclusions part so as to give more clarity to the summary information there. Regarding the working hypothesis, we have introduced it in the text of the manuscript, both in the Introduction part (in the form of conditional and deterministic elements, based on indicators/variables related to the evaluation of geosites, but also on image elements) and in the Conclusions, in the form of synthetic results related to the empirical part.
About the already published study on Yazd Province, partial study with quantitative focus, which we have not published (it is the study published by Omidvar and Khosravi, 2010), we are able to inform you that this study has already been used by us and is mentioned in the article. As you rightly mention, this study refers to the interest in geotourism through the attraction of different geosites. However, we should mention that our study takes into consideration in most cases a completely different list of attractions in Yazd Province. In addition, we are conducting an assessment of the potential for geotourism, which is not done in the other study. It is also worth mentioning that we also look at the determining and conditioning relationship between geotourism and ecotourism resources and landscapes as tourism resources and final element of interest for interviewees and visitors. A mention of the same study is that it confirms the same trend that we have already mentioned, namely that there is a tendency for geotourism and ecotourism to be considered thematically close, with frequent references in the study to both geotourism and ecotourism.
The authors do not yet cite the main article where Pralong states the method used here. The Pralong article to be included is: https://doi.org/10.4000/geomorphologie.350
Answer. Thank you for your reminder. Our haste made us forget the exact paper that gives the method applied in this study.
This reviewer acknowledges the efforts made by the authors to improve all figures in accordance with the recommendations offered in the first review. In relation to figure 1, the authors should unify the representation of the altitudes without distinguishing between Shirkooh and Ardakan, and Anjir.
Answer. Done
When developing the graphic scale of the figures contained in the maps, the authors refer to them sometimes in miles and sometimes in kilometers. Once they decide whether to use miles or kilometers, they should use the internationally accepted abbreviation mi or km on the graphic scale and not miles or kilometers.
Answer. Done
This manuscript is a resubmission of an earlier submission. The following is a list of the peer review reports and author responses from that submission.
Round 1
Reviewer 1 Report
Dear Authors,
The article is informative and well written . However, I do have a few comments:
1) Annotation should be changed:
- It should be a little bit shorter
- now it's very broad and too general
- it's too descriptive
- It has to be strictly focused on the main objectives, methodology, and results/conclusions.
The first half of the abstract should be deleted, replaced with a single sentence, and the second half should be significantly improved before being written.
2) Frequently, eco-tourists seeking to explore in detail the anthropo-cultural and historical resources of rural tourism take the opportunity to expand their knowledge of the petrographic and geological heritage at an appreciable distance from the place where they are staying [12,13] i.e. also by geotourism.
What do you mean by this phrase? This sentence is unclear.
3) There are deviations in the serial number of links. For example, line 6 [51], line 10, line 22 [31], etc.
4) Pictures are not allowed in most magazines, if the author does not have personal photographs of this object (Figures 2 and 5)
5) The conclusion needs to be rewritten. Figure 5 and its description in the conclusion must be in section 3.

Reviewer 2 Report
Please find my comments in the revised form of your manuscript. There is a long way to improve the quality of the paper

Reviewer 3 Report
A BRIEF SUMMARY
The present paper performs an evaluation of the landscapes of the Yazd-Ardakan Plain by applying the method proposed by Pralong in 2005 to evaluate the use and tourist potential of places of geomorphological interest. It is a well written and structured work, which is based on applying a good method (Pralong's method, 2005), but it is not an original method of the authors. The authors also do not propose a modification that improves it. The only contribution, although it is not original either, is the incorporation of an analysis of visual elements of the landscape based on Jakle (1987) and Bell (2004 and 2019). One of the most interesting results is the proposal made by the authors in section 5.1. where the thematic activities and the infrastructures that should be developed in the future to promote geotourism in the studied area are exposed.
GENERAL CONCEPTS COMMENTS
This paper is not a research article, nor is it a review article in the strict sense, although it seems more like a review article. There is no hypothesis to prove, nor is a proper method proposed to identify and evaluate the tourist interest of geomorphological sites. The authors apply methods proposed by other researchers, but without critically evaluating their usefulness, their successes, and their errors. The territory where the work is applied is not excessively relevant at the international level, in fact, most of the geomorphological sites evaluated present average values according to the method applied. Therefore, the work does seem very relevant at the national level but not at the international level. The references used are suitable and updated.
The use of the concept of ecotourism in the article is not sufficiently justified in the introduction or in the research background. The methods adopted are to evaluate the geotourism interest and to analyze the landscapes visually, but no methods are applied to evaluate ecotourism. On the other hand, this referee wonders if geotourism is not a part of ecotourism. For all these reasons, the elimination of the concept and the term ecotourism in the article is proposed.
The conclusions are a mixture of a summary of the article and personal reflections of the authors, since they do not really respond to validating any hypothesis, since the article does not present any working hypothesis. It is a work based on the application of methods, not designed by the authors, to evaluate the geotourism potential of a specific territory of Iran. It should be noted that the authors already published in 2010 a work on the interest for geotourism in the province of Yazd.
All these assessments of the referee should be considered when publishing the article.
SPECIFIC COMMENTS
The paper is very well written and formally very well presented and documented. With few spelling corrections to make.
Here are some modifications that authors should consider:
Page 2, 3rd paragraph, 2nd sentence: Geotourism is not only developed in natural areas, it can also be carried out in urban areas (see, for example, references 18 and 36 cited by the authors themselves).
The subsections of point 3.2. they should show more information about the characteristics of these landscapes, especially in the case of section 3.2.2. Siah-kooh. In this case, characteristics of the desert or the protected area are not mentioned. The species of flora and fauna commented on in the conclusions should be included in these subsections.
Section 4 dedicated to describing the Pralong Method seems excessive since the method is extensively exposed in the author's reference work: Jean-Pierre Pralong, «A method for assessing tourist potential and use of geomorphological sites», Géomorphologie: relief, processus, environment [Online], vol. 11 - No. 3 | 2005, online on October 1, 2007, consulted on November 8, 2022. URL http://journals.openedition.org/geomorphologie/350 ; DOI: https://doi.org/10.4000/geomorphologie.350. Consequently, the authors could refer to that publication to learn about the method and its application. It is surprising that the authors do not include this work in the references of the article, and it should be included.
Line 112: replace Perlong with Pralong
Line 220: You must include the scientific names of the animals cited (Persian cheetah and bustard)
In relation to the figures, it should be noted that they all require modifications to improve their visualization and understanding:
Figure 1: requires an important modification so that map C must be the protagonist and the one that occupies most of the space. A and B should occupy less space and should only serve to locate the study area. Map B should show the area represented on map C. The altitudes (no Heights) of the two referenced areas should be homogenized, there should only be a shared legend instead of a legend for each area. The legend of the ecotourism and geotourism attractions must be larger, as they are not legible.
Figure 2: Photos should be sorted based on how geotourism and ecotourism attractions are presented in the text or text should be sorted based on how photos are presented in the figure. A description of what the figure generally represents must be included, and then indicate what each of the photographs represents.
Figure 3: If Figure 1 were well done, it would be unnecessary to include the small orthoimage of this figure. Another improvement would be to avoid crossing arrows to represent the photos in their geographical position. To this end, it is proposed to reorder the photographs from A to P, avoiding crossing arrows as much as possible. The font size of the caption should be larger.
Figure 4: We are dealing with a figure that is an important contribution of the authors' results but requires a design improvement because it is unreadable. The first improvement is to enlarge the size of the figure, occupying an entire page if necessary. For this it is recommended to remove the legends from the left side and put them below all the maps. The legends of Geology and Environments and protected areas are incompatible with each other. An improvement should be made, such as using fewer pastel patterns and colors for Environments and protected areas. On the other hand, the symbols of hydrography and infrastructures for geo and ecotourism are too many to represent on a single map and their sizes are too small for reading. Some elements to represent should be grouped (for example: railway and roads, hotel, motel, ecoledges and villages, interpretation panels and viewpoints, information centers and museums, Pir-e naraki and chak chak) and the size of the symbols.